# Failure of Non-Invasive Respiratory Support in Patients with SARS-CoV-2

**DOI:** 10.3390/jcm12206537

**Published:** 2023-10-15

**Authors:** Juan Javier García-Fernández, José Andrés Sánchez-Nicolás, Sonia Galicia-Puyol, Isabel Gil-Rosa, Juan José Guerras-Conesa, Enrique Bernal-Morell, César Cinesi-Gómez

**Affiliations:** 1Health Sciences PhD Program, Catholic University of Murcia UCAM, Campus de los Jerónimos nº135, Guadalupe, 30107 Murcia, Spain; juanj2411@gmail.com; 2Emergency Department, Reina Sofía General University Hospital, 30003 Murcia, Spaincesarcinesi@gmail.com (C.C.-G.); 3Emergency Department, Los Arcos del Mar Menor General University Hospital, 30739 Murcia, Spain; 4Emergency Department, Rafael Méndez Hospital, 30813 Murcia, Spain; 5Infectious Diseases Department, Reina Sofía General University Hospital, 30003 Murcia, Spain

**Keywords:** non-invasive respiratory support, SARS-CoV-2, hospitalization, CPAP, HFNO, bilevel

## Abstract

Introduction: The objective of this study is to assess the failure of therapies with HFNO (high-flow nasal oxygen), CPAP, Bilevel, or combined therapy in patients with hypoxemic acute respiratory failure due to SARS-CoV-2 during their hospitalization. Methods: This was a retrospective and observational study of SARS-CoV-2-positive patients who required non-invasive respiratory support (NIRS) at the Reina Sofía General University Hospital of Murcia between March 2020 and May 2021. Results: Of 7355 patients, 197 (11.8%) were included; 95 of them failed this therapy (48.3%). We found that during hospitalization in the ward, the combined therapy of HFNO and CPAP had an overall lower failure rate and the highest treatment with Bilevel (*p* = 0.005). In the comparison of failure in therapy without two levels of airway pressure, HFNO, CPAP, and combined therapy of HFNO with CPAP, (35.6% of patients) presented with 24.2% failure, compared to those who had two levels of pressure with Bilevel and combined therapy of HFNO with Bilevel (64.4% of patients), with 75.8% associated failure (OR: 0, 374; CI 95%: 0.203–0.688. *p* = 0.001). Conclusions: The use of NIRS during conventional hospitalization is safe and effective in patients with respiratory failure secondary to SARS-CoV-2 infection. The therapeutic strategy of Bilevel increases the probability of failure, with the combined therapy strategy of CPAP and HFNO being the most promising option.

## 1. Introduction

Acute respiratory failure caused by SARS-CoV-2 emerged at the end of 2019 in China, becoming a pandemic within a few months [1]. The manifestations of SARS-CoV-2 vary depending on the patient’s condition, genetic variability, and viral strain [2]. Clinical presentations range from asymptomatic states to hypoxemic respiratory failure [3], which requires a wide therapeutic range, from conventional oxygen therapy through non-invasive respiratory support (NIRS) and invasive mechanical ventilation (IMV) to ECMO (extracorporeal membrane oxygenation) [4]. Bilateral pneumonias are characteristic of SARS-CoV-2, which generate, in serious states, acute respiratory distress syndrome (ARDS). Other authors have classified it as a new entity called acute respiratory distress syndrome caused by SARS-CoV-2 (C-ARDS) based on certain pathophysiological differences and divided it into two phenotypes (phenotype-L and phenotype-H) [5]. In addition, we must remember the “happy hypoxemia,” which refers to patients with objectively measured hypoxemia but with little or no clinical symptoms. A final consensus on the pathophysiological mechanism has not been reached, but its association with the H-phenotype of C-ARDS has been demonstrated [6]. However, other authors have proposed that this state of “happy hypoxemia” is caused by intrapulmonary shunts called acute vascular distress syndrome (AVDS) [7,8]. This tends to prolong the onset of symptoms, hence the importance of its management in the hospitalization ward [9,10].

Given the expeditious spread of the SARS-CoV-2 pandemic [11], hospital intensive care units (ICUs) were quickly filled, necessitating the application of non-invasive respiratory support therapies (NIRSs) in the hospital ward. These include high-flow therapies with nasal cannulas (HFNO), non-invasive mechanical ventilation (NIV) with either continuous positive airway pressure (CPAP), two levels of airway pressure (Bilevel), or a combination therapy between HFNO and NIV, regardless of the ventilatory mode. There is debate regarding which NIRS modality generates less failure. In previous studies that used NIRS, the failure rate was around 50% [12,13,14,15]. The therapy with the greatest failure is Bilevel, but there is controversy as to whether CPAP and high-flow therapy are superior to each other [16,17,18]. Although combination therapy is an increasingly used strategy in clinical practice, there is little literature on its use during the pandemic as well as its efficacy [19]. The objective of our study was to assess the failure of different NIRS therapies (HFNO, CPAP, Bilevel, or combined therapy) in patients with hypoxemic acute respiratory failure due to SARS-CoV-2.

## 2. Materials and Methods

An observational and retrospective study was designed for patients who came to the emergency room through the respiratory circuit for suspicious symptoms of SARS-CoV-2 at the Reina Sofía General University Hospital (RSGUH) of Murcia. The RSGUH attends an average of 93,000 emergencies per year, has 350 beds, and a reference population of 250,000 people. The study covered the period from 8 March 2020 to 26 May 2021. The inclusion criteria were SARS-CoV-2-positive patients who required NIRS during their hospital stay. Patients who required IMV prior to initiation of NIRS were excluded.

NIRS failure was defined as death or the initiation of IMV during admission. The decision to start NIRS was the responsibility of the physician, as well as the decision to discontinue, change, combine, or initiate the IMV. The parameters for the ventilator and medication were set at the physician’s discretion. However, as of April 2020, an institutional protocol was established to advise the start of NIRS in patients with PaO_2_/FiO_2_ < 200 mmHg and the need to use FiO_2_ greater than 40% to maintain adequate oxygenation or respiratory rate above 24 rpm. This protocol recommended the use of NIV in CPAP mode or HFNO as a first-line therapy.

This study followed the current laws and regulations and was approved by the Clinical Research Ethics Committee of RSGUH and the Catholic University of San Antonio (UCAM). Protocol number of the Ethics Committee: CE042207.

During the study, the following demographic variables were collected: age and sex; comorbidities: arterial hypertension (HTN), diabetes mellitus type 2, dyslipidemia, obesity, and chronic obstructive pulmonary disease (COPD); treatments: home oxygen therapy (HOT), continuous positive airway pressure (CPAP) at home, clinical values in triage (Glasgow and vital signs), biochemical, haematimetric, venous blood gas, D-dimer, and inflammatory biomarker data; dexamethasone treatment; place of onset of NIRS (hospitalization draw or emergency room); duration of NIRS (hours), hospital stay, and in-hospital mortality.

Qualitative variables were analyzed using absolute and relative frequencies. The quantitative variables were described by mean and standard deviation if they presented parametric distribution and by median and interquartile range in case of nonparametric distribution. The type of distribution was checked using the Kolmogorov–Smirnov test. Pearson’s chi-square or Fisher’s test was used to compare qualitative variables. Among the quantitative variables with normality criteria, the Student’s *t* test (in comparisons of one variable with two categories) or the analysis of variance (ANOVA) test (comparison of a variable with more than two categories) and those that did not meet normality criteria, the Mann–Whitney U or the Kruskal–Wallis tests, as appropriate. All significant factors were included in the univariate analysis. All analyses were performed in 2 tails, and statistical significance was accepted if *p* < 0.05 or 95% CI. The SPSS Statistics v-21 program (IBM, New Castle, NY, USA) was used.

## 3. Results

Of 7355 patients, 197 (11.6%) were included. Among the exclusions, 5666 (77%) patients were excluded because they were SARS-CoV-2-negative, 3 because of early orotracheal intubation (0.2%), and 1489 were SARS-CoV-2-positive patients (88%) who did not require NIRS. Of the 197 patients who required NIRS, 95 of them failed this therapy (48.3%) (Figure 1).

Table 1 shows the relationship between the variables studied and the type of NIRS. The median age of the patients treated with NIRS was 66 years, with an interquartile range (IQR) of 21; 129 (65.5%) of the patients were males. The most frequent personal history was hypertension (58.9%), diabetes mellitus type 2 (34%), dyslipidemia (44.7%), and obesity (22.3%). Further, 8.1% used home CPAP, and 5.1% used HOT. The SOFA of the patients on arrival at the emergency room was 2 (IQR 1) and SpO_2_/FiO_2_ 423 (IQR 52). Regarding the frequency of use of each type of NIRS therapy, we found that HFNO was used in 18 patients (9.1%), followed by combined HFNO therapy with CPAP in 24 patients (12.2%), CPAP was used in 28 patients (14.2%), combined HFNO therapy with Bilevel in a total of 45 patients (22.8%), and Bilevel in 82 patients (41.7%). The time of initiated NIRS from arrival at the emergency room was 24 (IQR 96) hours. Breaking down this result, the time of initiation of HFNO from arrival at the emergency room was 36 (IQR 93) hours, the combined therapies of HFNO with CPAP were 72 (IQR 60) hours, CPAP was 48 (IQR 99) hours, HFNO with Bilevel was 5 (IQR 95) hours, and Bilevel 24 (IQR 90) hours (*p* = 0.014). The median duration of the different respiratory therapies was 107 (IQR 164) hours, broken down by the type of NIRS: HFNO had a median duration of 48 (IQR 117) hours, combination therapy of HFNO with CPAP had a median duration of 96 (IQR 90) hours, CPAP had a median duration of 120 (IQR 120) hours, combination therapy with HFNO and Bilevel had a median duration of 156 (IQR 201) hours, and Bilevel a median duration of 96 (IQR 192) hours (*p* = 0.057).

In total, 55 (27.9%) patients required IMV. The percentages of need for IMV according to the type of therapy were 22.2%, 16.7%, 32.1%, 33.3%, and 28% for HFNO, CPAP + HFNO, CPAP, Bilevel + HFNO, and Bilevel, respectively (*p* = 0.611). Overall intrahospital mortality was 32.5% (64 patients), with therapy mortality of 16.7%, 4.2%, 21.4%, 37.8%, and 45.1% for HFNO, CPAP + HFNO, CPAP, Bilevel + HFNO, and Bilevel, respectively (*p* = 0.001).

Overall, the NIRS failure was 48.3% (95 patients). The failures according to the type of NIRS are shown in Table 2. Further, 70 patients (35.5%) received NIRS with HFNO, CPAP, or CPAP + HFNO, of which the technique failed in 23 patients (24.2%). The remaining 127 patients (64.5%) were treated with Bilevel or Bilevel + HFNO, with a failure of the technique in 72 patients (56.6%) (OR: 0.374; CI 95%: 0.203–0.688. *p* = 0.001). The analysis of NIRS failure in CPAP mode (46.5%) and HFNO (27.7%) showed no statistically significant relationship (OR: 2.253; CI 95%: 0.632–8.032. *p* = 0.206).

The variables associated with failure are shown in Table 3. Sex stands out, with a failure of 72.6% in men compared to 27.4% in women (OR: 1.858; CI 95%: 1.020–3.382. *p* = 0.042); the median age at failure was 69.8 (IQR 14.3) years versus success, which was 64.5 (IQR 12.2) years (*p* = 0.006) and, as antecedents, diabetes mellitus type 2, which was present in failure in 41.1% of patients versus 27.5% who were successful (*p* = 0.074). Upon arrival at the emergency room, the median SpO_2_/FiO_2_ in failure was 414 (IQR 90) versus success, which was 428 (IQR 33) (*p* = 0.013). The median lactic acid level at failure was 1.8 (IQR 1.3) mg/dL versus 1.4 (IQR 1.1) mg/dL (*p* = 0.001). The time of onset of NIRS in failure was 24 (IQR 96) hours, which was earlier than in success with an onset of 48 (IQR 95) hours (*p* = 0.758). The median duration of NIRS was 96 (IQR 168) hours at failure and at success was 126 (IQR 144) hours (*p* = 0.176). The median stay in the hospitalization ward was 12 (IQR 9.8) days for success, and failure was 10 (IQR 12) days (*p* = 0.016). NIRS was started in the hospitalization ward in 121 patients (61.4%) compared to 76 patients who started in the emergency room (38.6%), and NIRS failure that started in the ward was 51 patients (42.2%) compared to success in 70 patients (57.8%) (*p* = 0.031).

Regarding the distribution according to waves, the number of patients in the first, second, third, and fourth waves, respectively, was 6 (3%), 45 (22.8%), 129 (65.5%), and 17 (8.6%). The failure rates were 66.7%, 53.3%, 48.8%, and 23.5% in the first, second, third, and fourth waves, respectively (*p* = 0.141) (Figure 2).

## 4. Discussion

NIRS is a fundamental pillar in the treatment of respiratory failure secondary to SARS-CoV-2. The particularities of overloading the health system in the SARS-CoV-2 pandemic definitively opened the doors to the use of NIRS in conventional hospitalization wards.

In our study, we observed that the most frequently used therapy was Bilevel, followed by a combined Bilevel + HFNO therapy, with almost two-thirds of the patients receiving these treatments. However, when analyzing the strategies of CPAP and/or HFNO, as recommended by the consensus, they presented a markedly lower failure rate, being statistically significant. In this sense, it is important to highlight that the strategy with the least failures was combined therapy with CPAP + HFNO. A noteworthy finding in our data is the lower failure of NIRS when it starts in the conventional hospitalization ward with respect to starting in the emergency room. Finally, despite not being statistically significant, our study demonstrateded a lower failure rate in successive waves, probably because of the combination of a greater multidisciplinary approach and experience of the doctors at the hospital.

The consensus recommendations and guidelines on COVID-19 management (https://www.covid19treatmentguidelines.nih.gov/, access data 30 September 2023) lean towards HFNO or CPAP mode over Bilevel [20]. Caution is required when ventilating with two levels of pressure (Bilevel); therefore, using a support pressure is associated with a directly proportional relationship between the support pressure and tidal volume. It has been demonstrated that tidal volumes, mainly above 10 cc/kg ideal weight, have a higher probability of ventilator-induced lung injury (VILI) [21] and, therefore, a higher probability of failure. The problem with pressure ventilation in patients with spontaneous breathing, usually during NIV, is the impossibility of programming or limiting the tidal volume performed by the patient. That is, by not being able to control the tidal volume, which is the result of programmed support pressure and the patient’s own respiratory effort, there is a greater probability of failure of the VILI technique. For this reason, recommendations advise using a support pressure as low as possible; therefore, using HFNO and/or CPAP (without support pressure) is the most advisable modality [12,17,22].

The overall failure of the NIRS in other series varies between 40 and 60% [13,14,17,19], so our data are within the average of these margins. Focusing on mortality, the overall 32.5% was higher than the 26.6% reported by Franco et al., a study similar to ours, but lower than the 40.5% reported by Perkins et al. [14]. However, when we analyzed only the therapies of HFNO and/or CPAP in terms of the failure between the study by Franco et al. and ours, we observed very similar figures between both studies (47.7% vs. 46.5%), particularly for CPAP, but lower in our case when referring to HFNO (38% vs. 27.7%). Our study is pioneering in analyzing combination therapies; therefore, it is not possible to make a comparison between these therapies and other articles. However, it should be noted that the lowest failure rate (20.9%) was observed in the combined therapy between CPAP and HFNO. The reason for these results is difficult to determine. Probably, the fact that these were younger patients and the earlier start of therapy, which may have meant a later onset of respiratory failure, influenced the good results. However, the better tolerance of combined therapy, which entails longer CPAP times without requiring sedation or the need to remove the support due to patient intolerance, should not be ruled out as a key point for the greater success of the joint technique over individual ones [15].

If we assess the place of onset of the NIRS, we find that its onset in the conventional hospitalization ward (61%), in all its modalities, is more frequent with respect to the hospital emergency service (39%). This may be due to the syndrome of “happy hypoxemia” or a few advanced conditions upon arrival at the emergency room, which would cause the need for no initiation of NIRS. Thus, the greater failure of NIRS in emergency services (57.9%) could be caused by more severe conditions or more advanced disease and, therefore, by the later initiation of therapy, which requires the immediate initiation of NIRS. The fact that there is less failure of the NIRS in the conventional hospitalization ward indicates that it is possible to perform NIRS safely and effectively, mainly by providing adequate monitoring and training of health personnel. In this sense, it is important to highlight that at the end of the second wave, all patients with respiratory support were in the same conventional hospitalization ward, which included centralized monitoring and medical and nursing staff with more experience. It is difficult to measure the impact of these actions; however, from the third wave onwards, a greater number of successes than global failures was observed.

Among the limitations of this study, we found that, first, despite having an acceptable sample size, there was a substantial decrease when analyzed by group, especially in those treated purely with HFNO. Second, the NIV parameters were not assessed. Third, despite the existence of an institutional protocol, it only focused on non-combined therapies, NIV, or HFNO, so the times between the different types of NIRS in combination therapies differed in each patient. Fourth, the results obtained may have been because NIV was used in patients who, upon arrival at the emergency room, were more severe than in those who opted for other types of therapy. However, it is true that many of them were admitted with conventional oxygen therapy and later started the NIRS in the hospitalization ward due to a subsequent worsening of SpO_2_/FiO_2_. Fifth, the experience of the physician responsible for the management of NIRS could influence the success or failure of patients, either by prolonging HFNO over time instead of an earlier onset of NIV or due to an inadequate choice of modes and/or ventilatory parameters.

## 5. Conclusions

The use of NIRS during conventional hospitalization is safe and effective for patients with respiratory failure secondary to SARS-CoV-2 infection. Bilevel’s therapeutic strategy increases the probability of failure. The combined therapy strategy with CPAP and HFNO could be the most promising option.

## Figures and Tables

**Figure 1 jcm-12-06537-f001:**
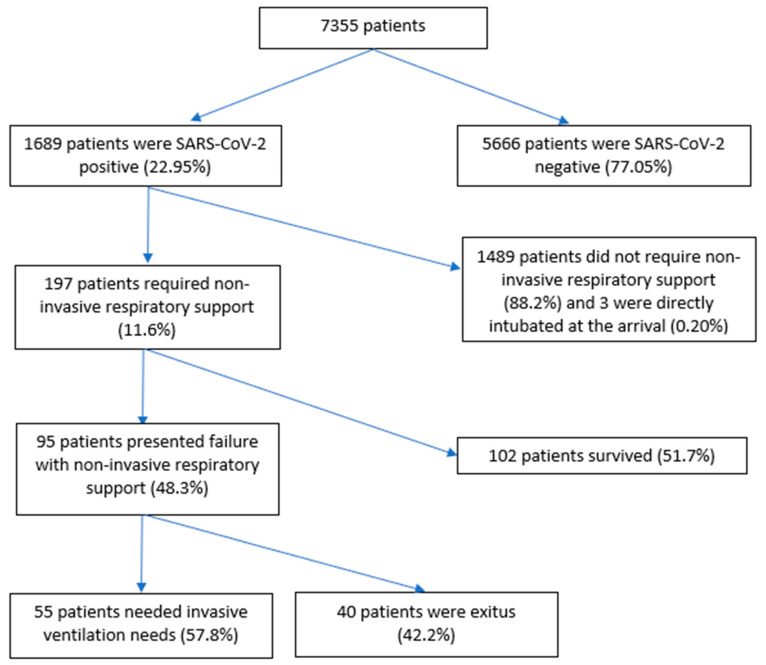
Flowchart of patient inclusion.

**Figure 2 jcm-12-06537-f002:**
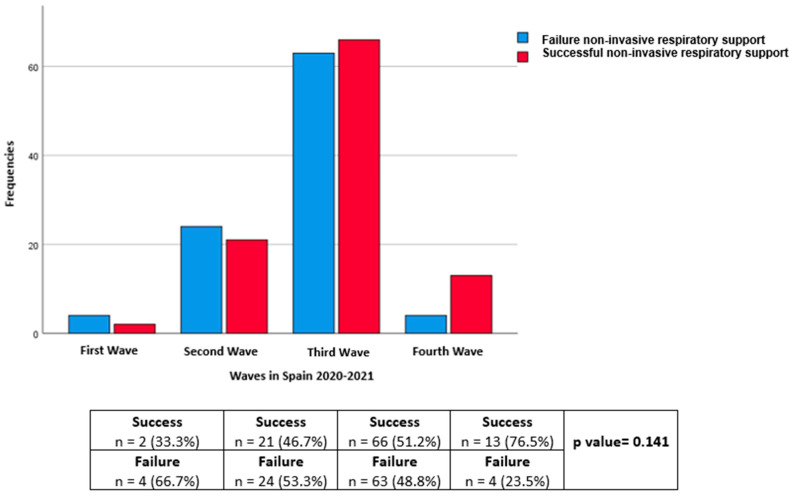
Success and failure by waves of SARS-CoV-2 in Spain.

**Table 1 jcm-12-06537-t001:** Clinical–analytical characteristics and evolution of the global sample and study according to the type of therapy used.

	Total N = 197 n (%)	HFNO N = 18 n (%)	CPAP + HFNO N = 24 n (%)	CPAP N = 28 n (%)	Bilevel + HFNO N = 45 n (%)	Bilevel N = 82 n (%)	*p* Value
Sex (male)	129 (65.5)	14 (77.8)	18 (75)	20 (71.4)	29 (64.4)	48 (58.5)	0.358
Age (years) **	66 (21)	61 (19)	62 (20)	61 (23)	63 (20)	73 (20)	0.001
HTN	116 (58.9)	11 (61.1)	11 (45.8)	12 (42.9)	30 (66.7)	52 (63.4)	0.164
Type 2 diabetes	67 (34)	5 (27.8)	5 (20.8)	9 (32.1)	20 (44.4)	28 (34.1)	0.355
Dyslipidemia	88 (44.7)	9 (50)	7 (29.2)	12 (42.9)	23 (51.1)	37 (45.1)	0.503
Obesity	44 (22.3)	4 (22.2)	6 (25)	4 (14.3)	11 (24.4)	19 (23.2)	0.863
HOT	10 (5.1)	0 (0)	0 (0)	0 (0)	0 (0)	10 (12.2)	0.005
CPAP home	16 (8.1)	0 (0)	2 (8.3)	1 (3.6)	6 (13.3)	7 (8.5)	0.402
COPD	12 (6.1)	0 (0)	0 (0)	2 (7.1)	2 (4.4)	8 (9.8)	0.296
MAP in triage (mm Hg) *	93 ± 14	97 ± 9.5	89 ± 14	95 ± 11	90 ± 15	93 ± 15	0.312
SatO_2_/FiO_2_ triage **	423 (52)	440 (63)	438 (36)	435 (52)	423 (57)	416 (68)	0.131
RR triage (1/min) **	26 (10)	26 (8)	21 (16)	25 (13)	25 (10)	26 (10)	0.508
Glucose (mg/dL) **	128 (70)	122 (72)	124 (36)	120 (122)	128 (101)	133 (57)	0.621
Creatinine (mg/dL) **	1.1 (0.5)	1.01 (0.4)	1.1 (0.3)	1.2 (0.5)	1.1 (0.6)	1.2 (0.6)	0.054
CRP (mmol/L) **	11 (14)	12 (10)	14 (14)	10 (8)	15 (14)	11 (14)	0.782
LDH (UI/L) **	378 (255)	358 (281)	407 (301)	331 (259)	432 (233)	355 (245)	0.350
IL-6 (Pg/mL) **	101 (112)	48 (80)	106 (71)	91 (113)	95 (123)	122 (122)	0.459
Procalcitonin (mg/dL) **	0.17 (0.33)	0.14 (0.1)	0.13 (0.28)	0.12 (0.33)	0.22 (0.52)	0.21 (48)	0.147
pH **	7.40 (0.09)	7.40 (0.07)	7.41 (0.09)	7.41 (0.07)	7.40 (0.09)	7.41 (0.11)	0.301
PaCO_2_ (mmHg) **	42 (12)	41 (14)	42 (9)	41 (11)	40 (15)	43 (15)	0.157
HCO_3_- (mmol/L) *	25 ± 4.4	23 ± 3.5	25 ± 3.3	24 ± 6.7	24 ± 3.6	25 ± 4.8	0.059
Lactic acid (mg/dL) **	1.6 (1.3)	2 (1.3)	1.5 (1.2)	1.6 (0.9)	1.8 (1.2)	1.6 (1.2)	0.323
D-dimer (Ng/mL) **	798 (866)	892 (1648)	702 (687)	727 (886)	776 (659)	824 (1135)	0.989
Chest X-ray RALE **	5 (4)	4 (3)	4 (4)	4.5 (4)	6 (5)	6 (5)	0.192
SOFA score (sepsis) **	2 (1)	2 (2)	2 (2)	3 (2)	2 (2)	3 (2)	0.102
Dexamethasone treatment (mg) **	6 (0)	6 (2)	6 (2)	6 (0)	6 (14)	6 (0)	0.435
NIRS start time since arrival (hours) **	24 (96)	36 (93)	72 (60)	48 (99)	5 (95)	24 (90)	0.014
Time with NIRS (hours) **	107 (164)	48 (117)	96 (90)	120 (120)	156 (201)	96 (192)	0.057
Time in ward hospital (days) **	11 (9)	12 (11.5)	12 (6.5)	12 (5.5)	9.5 (12.3)	11 (12.3)	0.821
IMV Requirements **	55 (27.9)	4 (22.2)	4 (16.7)	9 (32.1)	15 (33.3)	23 (28)	0.611
In-hospital mortality **	64 (32.5)	3 (16.7)	1 (4.2)	6 (21.4)	17 (37.8)	37 (45.1)	0.001
Start of NIRS in hospitalization ward **	121 (61.4)	12 (66.7)	23 (95.8)	20 (71.4)	22 (48.9)	44 (53.7)	0.001

* Results are expressed as mean (standard deviation). ** Results expressed as median (interquartile range). HTN, hypertension; HOT, Home oxygen therapy; COPD, chronic obstructive pulmonary disease; MAP, mean arterial pressure; SatO_2_, oxygen saturation; FiO_2_, inspired fraction of oxygen; RR, respiratory rate; CRP, C-reactive protein; LDH, lactate dehydrogenase; IL-6, interleukin-6; RALE: Radiographic Assessment of Lung Edema; SOFA: Sequential Organ Failure Assessment Score; HFNO, high-flow nasal oxygen; CPAP, continuous positive airway pressure; Bilevel: breathing support with two levels of pressure; NIRS, noninvasive respiratory support; IMV, invasive mechanical ventilation.

**Table 2 jcm-12-06537-t002:** Success or failure of patients according to the modality of NIRS.

	Success n (%)	Failure n (%)	Total n (%)	*p*-Value Comparing between Therapies	Overall *p*-Value
HFNO	13 (72.3)	5 (27.7)	18 (9.1)		0.005
				*p* = 0.601 ^a^	
Combination therapy (CPAP + HFNO)	19 (79.1)	5 (20.9)	24 (12.2)		
				*p* = 0.053 ^b^	
CPAP	15 (53.5)	13 (46.5)	28 (14.2)		
				*p* = 0.697 ^c^	
Combination therapy (Bilevel + HFNO)	22 (48.8)	23 (51.2)	45 (22.8)		
				*p* = 0.347 ^d^	
Bilevel	33 (40.3)	49 (59.7)	82 (41.7)		
Total according to NIRS	102	95	197	*p* = 0.014 ^e^	

^a^ Comparison between HFNO with CPAP + HFNO *p* = 0.601; ^b^ comparison between CPAP + HFNO with CPAP *p* = 0.053; ^c^ comparison between CPAP with Bilevel + HFNO *p* = 0.697; ^d^ comparison between Bilevel + HFNO with Bilevel *p* = 0.347; ^e^ comparison between Bilevel with HFNO *p* = 0.014. Overall *p*-value in the comparison between the different types of NIRS compared to the success or failure of NIRS.

**Table 3 jcm-12-06537-t003:** Clinical–analytical characteristics and univariate study depending on the success or failure of NIRS.

	Success N = 102 n (%)	Failure N = 95 n (%)	*p* Value
Sex (male)	60 (58.8)	69 (72.6)	0.042
Age (years) **	64.5 (12.2)	69.89 (14.3)	0.006
HTN	57 (56.4)	60 (62.1)	0.420
Type 2 diabetes	28 (27.5)	39 (41.1)	0.074
Dyslipidemia	39 (38.2)	49 (51.6)	0.060
Obesity	21 (20.5)	23 (24.2)	0.542
HOT	3 (2.9)	7 (7.4)	0.168
CPAP Home	7 (6.9)	9 (9.5)	0.503
COPD	6 (6.3)	6 (5.9)	0.899
MAP in triage (mm Hg) *	94 ± 14.5	92 ± 14.6	0.337
SatO_2_/FiO_2_ in triage **	428 (33)	414 (90)	0.013
RR in triage (1/min) **	26 (7)	30 (12)	0.062
Glucose (mg/dL) **	121 (58)	135 (85)	0.199
Creatinina (mg/dL) **	1.07 (0.43)	1.28 (0.64)	0.051
CRP (mmol/L) **	9.85 (14)	14.3 (13.8)	0.050
LDH (UI/L) **	358 (222)	426 (286)	0.191
IL-6 (Pg/mL) **	83 (88)	140 (151)	0.242
Procalcitonin (mg/dL) **	0.13 (0.26)	0.21 (0.51)	0.802
pH **	7.40 (0.07)	7.40 (0.10)	0.426
PaCO_2_ **	42 (12)	40 (14)	0.971
HCO_3_- (mmol/L) *	25 ± 4.6	25 ± 4.1	0.366
Lactic acid (mg/dL) **	1.4 (1.1)	1.8 (1.3)	0.001
D-dimer (Ng/mL) **	768 (751)	832 (1246)	0.488
Chest X-ray RALE **	4 (3)	6 (4)	0.002
SOFA score (sepsis) **	2 (2)	3 (2)	0.001
Dexamethasone treatment (mg) **	6 (2)	6 (0)	0.355
NIRS start time since arrival (hours) **	48 (95)	24 (96)	0.758
Time with NIRS (hours) **	126 (144)	96 (168)	0.176
Time in ward hospital (days) **	12 (9.8)	10 (12)	0.016
Start of NIRS in hospitalization ward	70 (57.8)	51 (42.2)	0.031

* Results are expressed as mean (standard deviation). ** Results expressed as median (interquartile range). HTN, hypertension; HOT, Home oxygen therapy; COPD, chronic obstructive pulmonary disease; MAP, mean arterial pressure; SatO_2_, oxygen saturation; FiO_2_, inspired fraction of oxygen; RR, respiratory rate; CRP, C-reactive protein; LDH, lactate dehydrogenase; IL-6, interleukin-6; RALE: Radiographic Assessment of Lung Edema; SOFA: Sequential Organ Failure Assessment Score; CPAP, continuous positive airway pressure; NIRS, noninvasive respiratory support.

## Data Availability

Data are unavailable due to privacy or ethical restrictions.

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
