# Peer review of "Failure of Non-Invasive Respiratory Support in Patients with SARS-CoV-2"

_jcm, 2023, doi:10.3390/jcm12206537_

Round 1

Reviewer 1 Report

Introduction section may be shortened.

References to the current clinical guidelines on COVID management should be mentioned in discussion section (https://www.covid19treatmentguidelines.nih.gov/).

Why "SatO2/FiO2" was used instead of PaO2/FiO2 ?

Author Response

Dear Reviewer, 

Thank you so much for your work and time. I answer you point by point here:

1) Introduction section may be shortened: I will do shorter introduction. Thank you.

2) References to the current clinical guidelines on COVID management should be mentioned in discussion section (https://www.covid19treatmentguidelines.nih.gov/). You are right. I will add in the text immediately.

3) Why "SatO2/FiO2" was used instead of PaO2/FiO2: We decided to use the SatO2/FiO2 because equivalence is proven* and because it was used as the first indicator and is faster to obtain than the PaO2/FiO2

* Todd W. Rice, Arthur P. Wheeler, Gordon R. Bernard, Douglas L. Hayden, David A. Schoenfeld, Lorraine B. Ware. Comparison of the Spo2/Fio2 Ratio and the Pao2/Fio2 Ratio in Patients With Acute Lung Injury or ARDS, Chest, Volume 132, Issue 2, 2007, Pages 410-417, ISSN 0012-3692, https://doi.org/10.1378/chest.07-0617. (https://www.sciencedirect.com/science/article/pii/S0012369215374316)

Miranda MC, López-Herce J, Martínez MC, Carrillo A. Relación de la relación PaO2/FiO2 y SatO2/FiO2 con la mortalidad y la duración de ingreso en niños criticamente enfermos [Relationship between PAO2/FIO2 and SATO2/FIO2 with mortality and duration of admission in critically ill children]. An Pediatr (Barc). 2012 Jan;76(1):16-22. Spanish. doi: 10.1016/j.anpedi.2011.06.006. Epub 2011 Aug 25. PMID: 21871849.

Appreciate your work correction. It will improve the quality of the text.

Thank you so much.

Juan Javier García Fernández.

Reviewer 2 Report

Thanks for the possibility to review this retrospective study investigating the use of NIV in SARS CoV2-infected patients in a Spain hospital. The authors included 197 patients presenting with severe hypoxemia. They concluded that amongst different strategies of non-invasive ventilation bilevel showed the highest rate of failure in terms of changing to invasive ventilation or death of the patients.

Despite being an interesting idea, several major issues need to be addressed by the authors:

Generally, a comparison of the used (mean) pressures of the different NOV strategies would be of interest. The author correctly stated this point in the limitations.

Second, the high rate of failure within patients with bilevel (either with HNFO or alone) raises the question if this was really attributable to the ventilation strategy itself. Or could it have been that bilevel was used with the sickest patients as these had the highest share of HOT and home CPAP? Is this a possible explanation for the high failure rate?

Or could there have been a difference in terms of used pressure levels? One could hypothesize that higher tidal volumes with bilevel could have led to more VILI.

Furthermore, please see the specific comments enclosed below:

Line 26ff: this sentence is hard to understand. As there are presented 2 results, may be splitting into 2 sentences?

Introduction:

Line 49ff: please make shorter sentences to increases readability

Line 53f: … "happy hypoxemia" which are patients with objectified hypoxemia, but with little or almost any symptoms.

Line 58ff: improve the English like: as this tends to prolong the onset of symptoms, …

Line 61ff: first of all, this sentence is too long. Please make 2-3 sentences.

Secondly, you could start like: given the expeditious spread, …

Methods:

Line 91ff: could you provide a protocol number of the Ethics Committee?

Line 98: what do you mean with “gasometric”? bold gas analysis??

Results:

Line 120: think of rephrasing like: 129 patients were males.

Line 127 and 129: either: The time of initiation of NIRS … or: the time of initiated NIRS … (the same for HFNO)

Line 129: please check the spelling carefully: RIQ = IQR can be found several times

Do you have an explanation why HFNO with bilevel was commenced so early (only after 5h)? Did these patients show an earlier hypoxic deterioration?

Line 139: again, please check the spelling carefully! Mortality was 32.5% (64 patients); Not the other way round

Flowchart 1: could you present the decedents separated from those needing invasive ventilation?

Table 1: rpm is a rather unusual unit for breathing rate in triage. What about 1/min?

Change Glycemia to the more precise glucose.

Table 2: this table very hard to understand, I think it should be re-designed. Explain every abbreviation, what is TAFCN?

Did you compare the failure rates of all NIRS strategies? Did bilevel really fail more often then CPAP?

Figure 2: check the spelling, third and fourth wave. Red quadrat: successful non-invasive ….

Discussion:

Line 256: probably the fact that … : “these” instead of “they”?

Line 268: the fact that in the conventional hospitalization ward there is less failure of the NIRS … Could there be a bias as all sicker patients could have already been picked out and sent to ICU?

Line 273: centralized monitoring and after training. I think “after” is superfluous  

Line 284: what do you mean with “in the plant started “?

Conclusions:

Line 291: don´t you think that the increased probability of failure with bilevel may not be attributed to the ventilation itself as more to the degree of sickness?

I suggest a thorough check of the phrasing and the spelling. Please see the specific comments provided above.

Author Response

Dear Reviewer,

Thank you so much for your correction. I am sure that it will improve the quality of the investigation. Appreciate your great work. I will answer you point by point and add the changes in the manuscript.

Round 2

Reviewer 2 Report

thanks for responsing to all of my questions, manuscript improved significantly and I therefore suggest acception.